# Comparative Effectiveness Research: A Roadmap to Sail the Seas of IBD Therapies

**DOI:** 10.3390/jcm11226717

**Published:** 2022-11-13

**Authors:** Daniela Pugliese, Sara Onali, Giuseppe Privitera, Alessandro Armuzzi, Claudio Papi

**Affiliations:** 1CEMAD, IBD CENTER, Unità Operativa Complessa di Medicina Interna e Gastroenterologia, Dipartimento di Scienze Mediche e Chirurgiche, Fondazione Policlinico Universitario “A. Gemelli” IRCCS, 00168 Rome, Italy; 2Department of Medical Science and Public Health, University of Cagliari, 09124 Cagliari, Italy; 3Dipartimento Universitario di Medicina e Chirurgia Traslazionale, Università Cattolica del Sacro Cuore, 00042 Rome, Italy; 4IBD Center, IRCCS Humanitas Research Hospital, 20089 Rozzano, Milan, Italy; 5Department of Biomedical Sciences, Humanitas University, 20090 Pieve Emanuele, Milan, Italy; 6IBD Unit, “San Filippo Neri” Hospital, 00135 Rome, Italy

**Keywords:** Crohn’s disease, ulcerative colitis, biologics, target therapy, head-to-head, meta-analysis, real-world

## Abstract

The drug pipeline for the treatment of inflammatory bowel disease (IBD) has dramatically expanded over the last two decades, and it is expected to further grow in the upcoming years with the introduction of new agents with different mechanisms of action. However, such an increase of therapeutic options needs to be paralleled with an appropriate development of research to help physicians in the decision-making process when choosing which drug to prescribe. On the population level, comparative effectiveness research (CER) is intended to explore and identify relevant differences—in terms of both efficacy and safety outcomes—amongst different therapeutic regimens and/or strategies, in order to find the correct placement for each treatment in the therapeutic algorithm. CER revolves around three cornerstones: network meta-analyses, head-to-head trials and real-world studies, each of which has specific pros and cons, and can therefore offer answers to different questions. In this review, we aim to provide an overview on the methodological features specific to each of these research approaches, as well as to illustrate the main findings coming from CER on IBD target therapies (i.e., biologics and small molecules) and to discuss their appropriate interpretation.

## 1. Introduction

The therapeutic armamentarium for the treatment of inflammatory bowel disease (IBD) has been considerably widening over the last two decades, and more agents with novel mechanisms of action are expected in the upcoming years [1]. However, this increasing number of drugs, while offering more opportunities for IBD patients to achieve disease control, increases the level of uncertainty that physicians experience when choosing which treatment to administer.

Most guidelines are not helpful in this decision process, as the Grading of Recommendations, Assessment, Development and Evaluation (GRADE) [2] system only allows for the appropriateness of a specific therapy to be established, without providing any indication on the therapeutic agents that should be preferred in regard to either effectiveness or safety. For instance, the European Crohn’s and Colitis Organization (ECCO) guidelines [3] on the management of ulcerative colitis (UC) do not express specific recommendations on which agent(s) should be preferably chosen to induce remission in steroid-dependent or -refractory patients. Conversely, the 2020 AGA guidelines suggest in favor of the use of infliximab and vedolizumab over adalimumab to induce remission in bionaïve UC patients [4], based on the 2020 network meta-analysis performed by Singh and colleagues [5]; the most recently published Italian Group for the Study of Inflammatory Bowel Disease (IG-IBD) [6] guidelines on the use of biological and small molecules for ulcerative colitis (UC) make conditional recommendations on the use of some agents over others in naïve patient, but they do not provide a comprehensive therapeutic algorithm.

Comparative effectiveness research (CER) is defined as “The generation and synthesis of evidence that compares the benefits and harms of alternative methods to prevent, diagnose, treat, and monitor a clinical condition or to improve the delivery of care (…). The purpose of CER is to assist consumers, clinicians, purchasers, and policy makers to make informed decisions that will improve health care at both the individual and population levels” [7]. Hence, CER aims to respond to two core questions: (1) what should I do for the patient that I am treating right now, considering the different options available? (2) at the population level, what is the correct positioning in the therapeutic algorithm—based on effectiveness, safety, and costs—of each different therapeutic agent? CER is founded on three cornerstones as sources of evidence: network meta-analyses, head-to-head trials (ideally, with a pragmatic design) and real-world (observational) studies, each with its own perks and flaws. With regard to outcomes, efficacy is used to define the “ability” of a drug to produce a pre-specified effect under strict investigation circumstances (i.e., in placebo-controlled-RCTs), while effectiveness is used to describe the overall beneficial effect of an intervention when employed in everyday clinical practice (i.e., at the population level). In that regard, it is important to note that effectiveness can also be assessed in RCTs with a pragmatic design, where protocols are less stringent, and the investigated intervention is usually compared to the standard of care [8]. The purpose of this review is to describe the methodological characteristics, with their respective strengths and weakness, of all these three areas of CER, in order to facilitate the appropriate interpretation of the results coming from different types of studies, with specific interest for IBD, by providing examples for each study design.

## 2. Network Meta-Analyses

Traditional meta-analysis is a statistic technique that allows the cumulation of evidence from different studies in order to report the overall effect estimate for a specific treatment: it consists of a pair-wise comparison between a certain therapeutic intervention and the same comparator (usually placebo), pooling data from different studies [9]. Meta-analyses aim at: (1) settling controversies when reports disagree, (2) increasing the statistical power of the hypothesis test, (3) refining the estimate of the effect (i.e., narrowing the confidence interval of effect size), and (4) highlighting the areas where evidence lacks. Meta-analyses are therefore intended to confirm the appropriateness of drug approval, substantiate the statements and recommendations coming from guidelines, and inform clinical decision making.

A network meta-analysis (NMA) can be considered a sort of “virtual head-to-head trial”, allowing for the measurement of the differential effect between two (or more) treatments. The simplest model of NMA is called “anchored indirect treatment comparison” (Figure 1A): if studies comparing agents A vs. C, and agents B vs. C are available, the NMA methodology will allow inference on the comparison between agents A and B, as there are data for both agents compared to a common comparator (in this example, agent C); in this case, the effect estimate of A vs. B is only indirect, as no direct comparison between the two agents is available. In a more complex model, called “mixed treatment comparison” (Figure 1B), direct evidence is available for multiple comparators (for instance, following on the example above, agents A vs. B, B vs. C, and A vs. C): in this case, the NMA methodology will allow for a combination of direct and indirect evidence to infer on the comparison between agents A and B [10].

NMAs are based on three assumptions: (1) similarity, i.e., the studies need to have similar methodologies, in terms of design, enrolled populations, outcome measures and effect modifiers, (2) homogeneity, i.e., results from pairwise comparisons (direct estimates) need to be coherent, (3) consistency and transitivity, i.e., direct and indirect evidence should not have major discrepancies or inconsistencies [10]. Nevertheless, such assumptions are considerably hard to respect, owing to the significant evolution in the design of clinical trials over the last decades [11]. Indeed, the first pivotal RCTs with biologics in IBD included two separate studies for the induction and maintenance phases. Subsequently, combined induction and maintenance trials were introduced, where patients were only randomized at the enrollment (i.e., treat-through design) and, in the most recent RCTs, responding patients (either all responders or only those who responded to the active treatment) are re-randomized at the end of induction—of note, these last designs also include the possibility of an open-label rescue arm for nonresponder patients. Furthermore, objective measures of disease activity are mandatory in contemporary RCTs, whereas the first trials were more commonly based only on subjective clinical activity scores. Another key issue to consider is that the patient population included in today’s RCTs is significantly different compared to the first studies—for instance, the latter only included bionaïve patients, whereas now at least half of enrolled patients have been previously exposed to at least one targeted therapy [12]. Hence, the comparison between different RCTs that span a 20-year period can be deceitful. Finally, RCTs are encumbered by a low external validity, as they are designed to maximize internal consistency, but the population enrolled does not necessarily reflect the actual IBD population encountered by physicians in their everyday clinical practice [12].

The results of NMA are usually presented either in the form of a table of consistency or as Surface Under the Cumulative Ranking Curve (SUCRA). Tables of consistency summarize the results of each comparison coming from direct and indirect evidence, expressed as Odds Ratio (OR) and Confidence Intervals (CIs), thus providing indications as to whether significant differences exist between the effects of two specific treatments. On the other hand, the SUCRA method expresses the probability that, among all the treatments assessed, a specific one is ranked first, second, third, etc., in regard to the predefined outcome: such a probability is expressed as a range (0 to 1, or 0% to 100%) associated with each treatment, with the highest values indicating a high probability that the treatment is in the top ranks and, conversely, the lowest values indicating that the treatment is more likely to be in bottom ranks for a certain outcome [13]. Nevertheless, the SUCRA can be misleading due to a variety of reasons: (1) it is affected by the quality of evidence based on which is calculated—hence, if it is poor, the SUCRA ranking is less reliable; (2) a specific treatment can be the best for a certain outcome, but also the worst for another one: for instance, agent A can be the best in terms of efficacy, but it can also rank as the worst in regard to safety; (3) SUCRA does not provide an estimation of the magnitude of difference in effects among different treatments.

Of note, in a 2013 paper it was reported that NMA present significant methodological issues: among 121 NMA analyzed, 30% did not clearly report the primary outcome, 50% did not report information on the risk of bias in individual RCTs, 73% did not report the methodology for the electronic research strategy used, 79% did not report the characteristics of the studies included, and 85% did not report the method used to assess publication bias [14]. Such data throw some shadows on the reliability of results coming from NMA and should warn in favor of caution when interpreting them. Finally, NMA should always include an evaluation on the quality of the included evidence, in order to appraise and “quantify” the confidence in effect estimates. In that regard, the GRADE system allows to provide a precise estimation, thereby downgrading the evidence from RCTs or, conversely, upgrading those from observational studies, based on pre-specified criteria [2].

### Network Metanalysis in IBD

Several NMAs have been published in the field of IBD, assessing many different outcomes in a variety of clinical scenarios (as summarized in Figure 2A,B). Here, we report the main findings coming from the two most recently published NMAs on the comparative efficacy and safety of biologics and small molecules in Crohn’s disease (CD) and UC.

The recent NMA by Singh and colleagues [15], assessing comparative efficacy and safety of targeted therapies for CD, included RCTs vs. placebo of infliximab, adalimumab, certolizumab pegol, vedolizumab, ustekinumab and risankizumab, together with a H2H trial comparing adalimumab vs. ustekinumab (SEAVUE) [16] and two H2H trials comparing infliximab or adalimumab alone vs. their combination with azathioprine (SONIC [17] and DIAMOND [18], respectively). The main findings are summarized in Table 1. Efficacy analysis for induction of clinical remission was performed separately for naïve and exposed patients. In regard to maintenance of clinical remission, only the EXTEND [19], SONIC [17] and DIAMOND [18] trials had a treat-through design; however, sensitivity analysis to adjust for differential trial designs did not show significant differences in outcomes. Overall, these data seemingly support the preferential use of the combination of infliximab + azathioprine for the treatment of bionaïve CD patients, as it appears to be associated with more favorable efficacy and safety profiles. In the exposed population, the role of vedolizumab might be questionable, and other agents should probably be preferred. Adalimumab appears to be an appropriate choice in exposed patients, but the evidence supporting its use are qualitatively poor. Indeed, there was low confidence in estimates supporting the use of adalimumab and vedolizumab over placebo in the exposed population, as the quality of evidence was downgraded due to the inclusion of specifically selected patients in the GAIN trial for adalimumab (only patients with loss of response or intolerance to infliximab, no primary nonresponders) [20] and the very serious imprecision with wide CI for vedolizumab. Notably, no data on endoscopic outcomes are reported in this NMA.

The NMA by Lasa and colleagues [21], assessing comparative efficacy and safety of targeted therapies for UC, included the registration trial vs. placebo of infliximab, adalimumab, golimumab, vedolizumab, ustekinumab, tofacitinib, upadacitinib, ozanimod and filgotinib, together with two registration trials vs. an active comparator or placebo (etrolizumab vs. adalimumab or placebo, HIBISCUS I and II [22]), and two H2H trials vs. an active comparator (adalimumab vs. vedolizumab, VARSITY [23], and etrolizumab vs. infliximab, GARDENIA [24]). The main findings are summarized in Table 2. Of note, the combination of infliximab + azathioprine was not assessed in this NMA. Among maintenance studies, the analyses were performed separately for the studies with a treat-through design (ev infliximab, sc infliximab, adalimumab, vedolizumab and etrolizumab) and those with responder re-randomization (golimumab, ev vedolizumab, sc vedolizumab, ustekinumab, tofacitinib, upadacitinib, filgotinib, etrolizumab and ozanimod). Overall, the data presented in this work suggest that upadacitinib might be considered the most effective agent to induce and maintain both clinical remission and endoscopic response, but it also appears to be the one with the least favorable safety profile. However, the absence of data on upadacitinib in naïve vs. exposed patients limits the definition of its positioning in the therapeutic algorithm. In naïve patients, infliximab appears to be the most effective choice to induce clinical remission. Vedolizumab is effective in inducing and maintaining both clinical remission and endoscopic response, mostly in naïve patients, with a good safety profile, thus representing a valid first-line therapy in at-risk populations; conversely, its role in exposed patients might be debatable. Tofacitinib and ustekinumab might be the most effective choice to induce both clinical remission and endoscopic improvement in the exposed population. Finally, filgotinib 100 mg does not appear to have any significant effect in the treatment of UC.

It is worth mentioning that both NMAs, including only data from RCTs, suffer from the same limitations as clinical trials with regard to the populations of patients included, thus limiting the generalizability of the data in everyday clinical practice. Specifically, CD patients <18 years of age, with mild disease activity or stricturing/penetrating disease behavior were not included. Likewise, for UC, patients <18 years of age, with mild disease activity or isolated proctitis were not included.

## 3. Head-to-Head Trials in IBD

To date, drug authorization and its positioning by health authorities mainly relies on data from RCTs with a placebo arm. However, the use of placebo in registration clinical trials may represent an unethical choice and a potential loss of opportunity for patients who nowadays have multiple approved therapeutic options. In this setting, trials directly comparing efficacy between new and established therapies are pivotal in helping physicians and payers to determine the position of new therapies.

Currently, head-to-head (H2H) trials represent the gold standard for CER, as they provide data as robust as those from RCTs, but also offer the possibility to directly compare drug efficacy (i.e., Varsity [23] AND SEAVUE [16] trial) or treatment strategies (e.g., REACT trial [25]) [26]. In addition, regulatory health agencies increasingly require an active comparator, not only the placebo control group, to establish the reimbursement of the new drugs. Nevertheless, it should be acknowledged that—in analogy with what already observed in regard to NMAs—the strict inclusion criteria of clinical trials prevent the generalizability of the results to the overall IBD population (i.e., low external validity).

The design of the H2H trial must satisfy some requirements. Different design of H2H trials exit with different strengths and limitations; the specific design mainly depends on the type and choice of comparator and the use of the blinding [27]. Based on the type of comparator, three different types of H2H trial design can be identified (examples of different trial designs are presented below and summarized in Table 3).

### 3.1. Placebo-Controlled Trial with a Non-Powered Reference Arm

In this type of trial, the active comparator serves to estimate the reference drug’s efficacy and to confirm previous results. This design offers some advantages, such as the small sample size needed, but on the other hand no solid conclusion can be drawn owing to the underpowered reference arm. Two trials with this design can be found for IBD. In 2007, Kamm et al. published the results of a trial comparing the efficacy of mesalamine multi-matrix (MMX), regular mesalamine (5-ASA), and a placebo in active UC patients [27,28]. In this trial, three treatments (two regimen doses of mesalamine MXX and standard 5-ASA) were compared, though the trial was powered to demonstrate the superiority of Mesalamine MMX over placebo and did not evaluate Mesalamine MMX over standard 5ASA.

Similarly, in 2012, active UC patients were randomized to receive budesonide MMX 6 mg/day, budesonide MMX 9 mg/day, 5-ASA 2.4 g/day, or placebo [29]. Again, the study was underpowered to demonstrate differences between budesonide MMX and 5-ASA.

In both studies, the authors were unable to draw solid conclusions about comparative efficacy to be used in clinical practice for drug positioning. These examples highlight the difficulty of, and the grade of uncertainty in, establishing firm conclusions based on trials in which the active comparator arm is underpowered.

### 3.2. Non-Inferiority Trials

Non-inferiority trials are designed to test if a new drug is non-inferior to one used as reference. In recent years, the use of this design exponentially increased, especially in trials sponsored by pharmaceutical companies. Non-inferiority trials aim to demonstrate a minimum level of efficacy of the tested drug, determined by the non-inferiority margin, corresponding to “the maximum difference clinically acceptable between two treatments”. The identification of this non-inferiority margin influences the sample size calculation. In IBD, this margin, which reflects the “clinically relevance”, is considered acceptable when the difference in efficacy between the investigational drug and the comparator is approximately 10% [33].

The identification of the correct comparator also represents a potential limit of these studies. Recently, Tsui et al., reviewed 162 non-inferiority trials published in high-impact medical journals in order to identify errors in the choice of the comparator. Only 25 out of 162 trials evaluated (15%) were correctly designed to compare an active treatment vs. an active comparator: 101 studies used, as the active comparator, a drug that previously had not even proven effective. Therefore, the results of these trials cannot be considered conclusive [34].

One of the principal examples of a non–inferiority design trial is “The Nor-switch trial” [31]. This is a non-inferiority, double blind, randomized trial investigating the efficacy of switching from infliximab originator to its biosimilar (CT-P13) in different autoimmune diseases. In this study the non-inferiority margin was set at 15%. Despite the achievement of the primary endpoint in the overall population, the validity of the data extrapolated from disease-specific subgroup analysis was impaired by the large confidence intervals exceeding the established non-inferiority margin.

### 3.3. Superiority Trials

This type of design represents the most solid strategy for determining the superiority of the investigated intervention over the comparator. In this setting, the calculation of the sample size provides the statistical power to demonstrate the relevant confidence limit for a difference that excludes zero and assumes that the effect of the interventional treatment is greater than the comparator of a predetermined “delta”. This type of test is more difficult to design, and the choice of the comparator becomes fundamental in order to correctly interpret the results. In H2H superiority trials, the comparator should be used in the best performing conditions. For instance, infliximab combined with immunosuppressor should be used as comparator instead of infliximab monotherapy [16], when using infliximab as a comparator. Indeed, in the IBD trial setting, the management of concomitant steroids and their tapering represent critical issues.

Currently, three H2H studies with a superiority design have been published in IBD. The VARSITY trial is the first H2H trial published in 2019, comparing two biological therapies in UC. Clinical remission was observed in a higher percentage in the vedolizumab than in the adalimumab group. However, these data should be interpreted with caution. Indeed, one should take into account that the study protocol did not allow the optimization of the two drugs, as opposed to what normally occurs in clinical practice, and neither an indication to stop steroids nor a steroid tapering protocol were indicated. Accordingly, the superiority of vedolizumab was not confirmed when considering steroid-free remission as endpoint [23].

More recently, two other H2H trials in IBD have been published. The SEAVUE trial was the first H2H trial powered to evaluate ustekinumab superiority over adalimumab for clinical remission at week 52, in moderate-to-severely active CD patients. In this trial, ustekinumab was not superior to adalimumab at week 52. Interestingly, the efficacy showed by both treatments were higher compared to previously published trials, and this could be explained by the absence of placebo control group, by the relative higher number of naïve patients and the shorter disease duration. These factors should be taken in account to translate these comparative data in clinical practice [16]. Finally, it is worth noting that the missing superiority of ustekinumab over adalimumab in moderate to severe CD, based on the superiority design of the SEAVUE trial, does not formally allow to conclude in favor of equal efficacy.

In the GARDENIA trial, comparing etrolizumab to infliximab in moderate-to-severely active UC patients naïve to TNF alpha inhibitors, the primary endpoint was not reached [24]. In addition to the choice of using, as the active comparator, infliximab monotherapy, instead of its combination with an immunosuppressor (which represents the best performing condition), another relevant aspect to be considered is that, similarly to the SEAVUE trial, the absence of superiority should not be interpreted as equal efficacy. Other than the evaluation of the comparator, further elements to consider to better understand the H2H trials results are the use of the single vs. double blinding, the choice of the endpoints, the dose, and the possible exit strategies.

In prospective comparative research, the correct use of the blinding is mandatory: indeed, the blinding strategy reduces the bias introduced by investigators and improves patient’s compliance. For regulatory purposes in which active control arm is required, double blinding is mandatory. Instead, if the aim of the trial is to demonstrate the comparative effectiveness in a real-world setting, single blind might be sufficient. The CONSTRUCT trial is an example of this type of design. This is a comparative study, evaluating the effectiveness of infliximab and cyclosporine in UC severely active patients. This is a non-inferiority single-blinded H2H trial in which the primary endpoint was quality-adjusted survival. A dynamic algorithm was used to generate allocations and protect against investigator preference. Local investigators and participants were aware of the treatment [30].

Currently there are numerous ongoing comparative studies on the efficacy of biological therapies with different mechanisms of action in IBD. Most of them present a double-blind double-dummy design and composite primary endpoints, as request from regulatory agencies. (See Table 4).

## 4. Real-Word Studies

Real-world studies (RWS) are positioned at the lowest step of the classical pyramid of evidence, as opposed to systematic reviews with meta-analyses and RCTs [35]. Under this general definition, several types of studies with different designs are included, such as cohort studies, case–control and cross-sectional studies, and case series studies. Accordingly, the quality of evidence coming from each study and the real impact on clinical practice can be quite variable. The ideal RWS should include a large cohort of patients (ideally >1000 across several sites) and adopt a propensity score (PS) matching to mimic randomization. The PS matching is a statistical method used to construct artificially homogenous groups for predefined covariates, in order to overcome the issue of selection bias and, therefore, obtain a more precise estimate of the effect of a specific drug or intervention. As a matter of fact, in clinical practice, patients are allocated to a specific treatment based on physicians’ judgment and awareness of an enhanced safety or effectiveness associated with a specific drug. Accordingly, especially in retrospective studies, comparisons between two treatment groups could be severely influenced by a high grade of heterogeneity, such as in terms of patients’ age, disease features, previous exposure to other drugs and comorbidities.

However, specific features of RWS confer them a fundamental scientific value. First, they include a significantly wider range of population. As we know, RCTs have strict inclusion and exclusion criteria, regarding disease characteristics (e.g., patients with isolated proctitis or with a history of a subtotal colectomy are commonly considered not eligible), prohibited drugs, predefined washout periods from previous medications (up to 12 weeks for vedolizumab in some clinical trials) and comorbidities. In 2012, Ha et al. quantified that about 31% of patients attending their referral IBD center would not quality for a pivotal RCT with biologics [13]. Apart from a previous exposure to biologics, which is an exclusion criterion that is not encountered anymore in recent trials, they identified that several other conditions, such as a structuring or a penetrating behavior (62%), presence of certain comorbidities (26%) and concurrent high-dose steroids exposure (18%) for CD patients and concomitant topical therapies (57%), a steroid or immunosuppressants naïve status (45%) or a new diagnosis (17%) for UC ones represented the most frequent reasons for the ineligibility. On the contrary, it is a common experience, in clinical practice, that patients start a targeted therapy while they are taking more than 30 mg of prednisone or after a few months from diagnosis, or without a previous exposure to steroids (i.e., top-down strategies). A history of malignancy, except for non-melanoma skin cancers, even after 5 years from the diagnosis, still limits the access to trials with targeted therapies. Conversely, several real-life data have been reported so far including patients exposed to biologics regardless of their previous neoplastic history, after an adequate counselling with oncologists and a careful consideration of the balance between risks and benefits associated with each choice [36]. Secondly, RWS allow for the potential exploration of long-term “hard outcomes”, including rates of surgery, hospitalization or access to emergency departments for IBD, and healthcare costs [37]. Thirdly, real-life data can serve for the extrapolation of effectiveness and/or safety data of a new drug from other indications. An example of that is represented by the experience with ustekinumab obtained by treating psoriasis and paradoxical forms in patients with IBD before its formal market licensing for gastroenterological indications [38,39].

RWS are also useful for the analysis of drug effectiveness and/or safety on specific aspects of disease (e.g., perianal disease, extra-intestinal manifestations) and on special population (e.g., pregnant and breastfeeding women, pediatric or elderly patients [40,41,42,43]). Finally, regarding comparison research, RWS often overcome the absence of head-to-head trials, potentially helping physicians to choose the best option for each patient. On the other side, lots of RWS have major limitations, for instance small sample size, lack of a control group, enrolment of a loosely selected population, wide variability in follow-up intervals and heterogeneity of outcome measures. Accordingly, data coming from studies with similar characteristics should be interpreted with caution, being encumbered by multiple biases. Table 5 provides a list of comparative RWS in IBD. 

### 4.1. Anti-TNF-α within the Class

Infliximab and adalimumab are licensed for the treatment of both CD and UC and golimumab only for UC. Currently, no specific recommendation exists on which is the best option for naïve patients. Even though some clinical characteristics (e.g., obesity or the presence of a complex perianal disease) could induce physicians to prefer infliximab, the choice is often determined by factors other than effectiveness, such as patients’ preference, the availability of an infusion center, or the insurance coverage in some countries.

In 2017, Singh et al. reported data extracted from the National Patient Registry and the Danish Drug Prescription Registry on CD patients treated with either infliximab or adalimumab as first-line biological therapy from 2005 to 2014; after a 2:1 PS matching on several variables, no significant differences were found in terms of CD major abdominal surgeries, new course of corticosteroids, IBD-related hospitalization, and serious infections. Conversely, a reduction in all-causes hospitalization was showed among patients treated with adalimumab [44]. This last finding is hardly interpretable, and the authors suggested the reason could be found in the study design, based on administrative claims and not on direct effectiveness measures.

Conversely, Macaluso et al. reported direct effectiveness comparison data on consecutive CD patients, split in two groups: a 2:1 PS-matched naïve group and a 1:1 experienced group. Rates of clinical benefit at 1 year were similar between infliximab and adalimumab treated patients in both groups and similar findings emerged also at 12 weeks. An increased rate of adverse events leading to drug withdrawal was reported overall among patients treated with infliximab, mainly related to infusion reactions [45].

The Danish group also reported data for a matched cohort of naïve UC patients [46]. Unlike CD, patients treated with adalimumab had an increased risk of all-causes hospitalizations and serious infections requiring hospitalizations, and an increased risk of all-cause and UC-related hospitalizations was reported in the subgroup of patients treated with combination therapy with immunomodulators plus adalimumab vs. plus infliximab. This superiority of infliximab over adalimumab could be partially related to the different pharmacokinetic profiles of these molecules (intravenous administration vs. subcutaneous, body weight-based vs. fixed dose). However, the aforementioned limits of the study design should be taken into account. Finally, a direct comparison between the two subcutaneous formulations of anti-TNF-α has been reported, showing a superiority of adalimumab over golimumab in terms of clinical benefit at week 8 and at the end of follow-up [47].

### 4.2. Anti TNF-α outside the Class (vs. Vedolizumab)

The Victory Consortium reported data on comparative safety and effectiveness between anti-TNF-α drugs and vedolizumab in both CD and UC adult patients using a multicenter PS–weighted cohort study. Regarding CD, no significant differences emerged between two groups in both risks of serious infections and serious adverse events. Conversely, the comparison on rates of non-infectious serious adverse events favored Vedolizumab over anti-TNF-α. This data is quite expected, since several class-specific adverse events, such as paradoxical psoriasis, hypersensitivity, or infusion reactions or drug-induced lupus are associated with TNFα inhibitors. With regard to effectiveness, despite treatment with anti-TNF-α drugs being associated with a higher persistence on therapy compared to vedolizumab, no significant differences emerged in terms of clinical remission), steroid-free clinical remission and endoscopic remission. However, vedolizumab was showed to be superior to subcutaneous anti-TNF-α drugs for both clinical outcomes among anti-TNF-α naïve patients. Disease duration seemed to significantly affect effectiveness, since an increased clinical benefit was recorded among patients with an early disease (≤2 years) treated with vedolizumab and on the contrary among patients with a late disease treated with anti-TNF-α drugs [48]. Moving to UC, the risk of serious adverse events and serious infections were similar between patients treated with vedolizumab and anti-TNF-α drugs, even though among anti-TNF-α naïve patients, treatment with vedolizumab seemed to be protective from serious adverse events. On the contrary, treatment with vedolizumab was associated with an increased risk of infections among anti-TNF-α exposed patients. Moving to effectiveness, Vedolizumab therapy appeared to be superior to anti-TNF-α treatment for inducing clinical remission, steroid-free clinical remission, and steroid-free deep remission. This gain was also maintained when analyzing separately anti-TNF-α naïve and exposed patients, and infliximab and subcutaneous anti-TNF-α-treated ones [49].

### 4.3. Vedolizumab vs. Ustekinumab in Crohn’s Disease

To date, six studies have been published comparing the effectiveness and safety of vedolizumab vs. ustekinumab for treating CD patients after failure of anti-TNF-α therapies, with inconclusive results [50,51,52,53,54,55]. Even though most of them seemed to reveal an overall superiority of ustekinumab over vedolizumab in terms of clinical effectiveness, discrepancies exist for specific outcomes, such as steroid-free clinical remission, at different timepoints. Moreover, except for the study by Bieman et al., also including biochemical remission among outcomes, the assessments of comparative effectiveness were only based on clinical activity.

Recently, Onali et al. published a collaborative study including the currently largest PS-matched cohort. At week 26, treatments with ustekinumab and vedolizumab were equal. Conversely, at week 52, treatment with vedolizumab was superior to ustekinumab in terms of clinical remission (OR 1.69, 95% CI 1.13–2.52) and steroid-free clinical remission (OR 1.53, 95% CI 1.02–2.28). The increased effectiveness of vedolizumab compared to ustekinumab was associated with younger age (<40 years old, OR 1.75, 95% CI 1.11–2.77), failure to two anti-TNF-α drugs (OR 2.55, 95% CI 1.34–4.85), absence of upper gastrointestinal involvement (OR 1.56, 95% CI 1.03–2.36), and a history of perianal disease (OR 4.21, 95% CI 1.10–16.09), concomitant immunosuppressive (OR 9.89, 95% CI 2.10–46.6) or steroids therapy (OR 2.33, 95% CI 1.18–4.62), and mild-to-severe disease activity at baseline (OR 1.53, 95% CI 1.02–2.28). A comparative evaluation of objective outcome measures was also performed, and no significant differences emerged between the two groups. Finally, safety was also comparable between the two groups [53].

## 5. Conclusions

In conclusion, NMAs, head-to-head trials, and real-world studies represent the foundations of CER, each with specific strengths and weaknesses (Figure 3). NMAs have the advantages of incorporating high quality evidence, of estimating a more precise effect, and can allow multiple comparisons and ranking. On the contrary, the validity of data can be seriously compromised by the quality of each study included, the research and publication bias, and the inconsistencies across the studies. Head-to-head trials represent the gold-standard of CER, for their strong internal validity and exclusion/control of confounding factors. However, they often suffer of a poor external validity, include a selected population not representative of real-life patients, and are quite expensive and time-consuming. Lastly, the robust external validity and the exploration of hard outcomes in the long-term represent the main pros of RWS. On the other hand, their low internal consistency, the heterogenicity of enrolled population, and the lack of uniformity in data completeness and accuracy represent the main pitfalls.

In this paper, we exclusively focused on biologics and small molecules, as their positioning is currently one the main topics in the IBD field. However, other therapies exist to both induce and maintain remission, and NMA and comparative studies are indeed important to compare the effectiveness and safety of corticosteroids [56], conventional therapies [57], and fecal transplantation [58,59] in IBD patients as well. Figure 4 summarizes the main findings from each methodology of CER. Despite providing some useful indication regarding the likely appropriate positioning of IBD therapies, further research is undoubtfully needed. Indeed, there is still a significant gap in available evidence on comparative research, which prevents physicians from having appropriate confidence in their decision when choosing which drug to prescribe. Methodologically sound H2H trials and RWS will be a priority in upcoming years, while new drugs are being introduced to the market, to ensure the best healthcare for IBD patients.

## Figures and Tables

**Figure 1 jcm-11-06717-f001:**
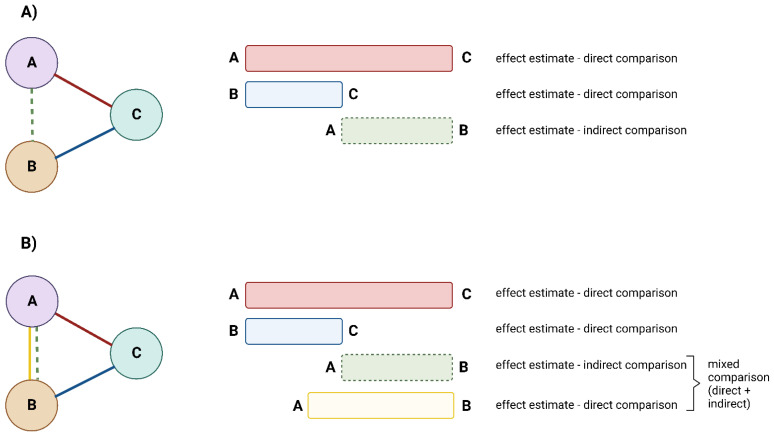
Network meta-analysis schematic. (**A**): Anchored indirect treatment comparison; (**B**): Mixed treatment comparison.

**Figure 2 jcm-11-06717-f002:**
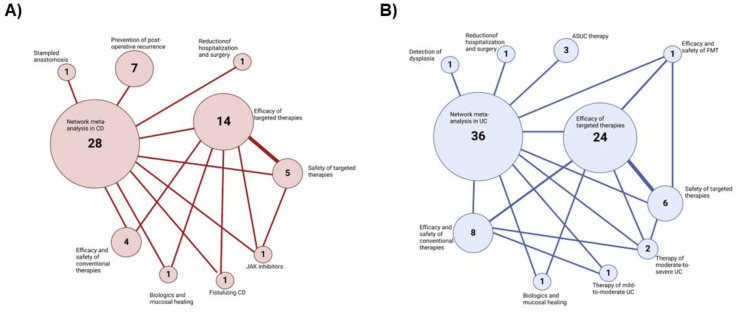
Overview of meta-analyses published in the field of IBD. (**A**):Meta-analyses in CD; (**B**):Meta-analyses in UC.

**Figure 3 jcm-11-06717-f003:**
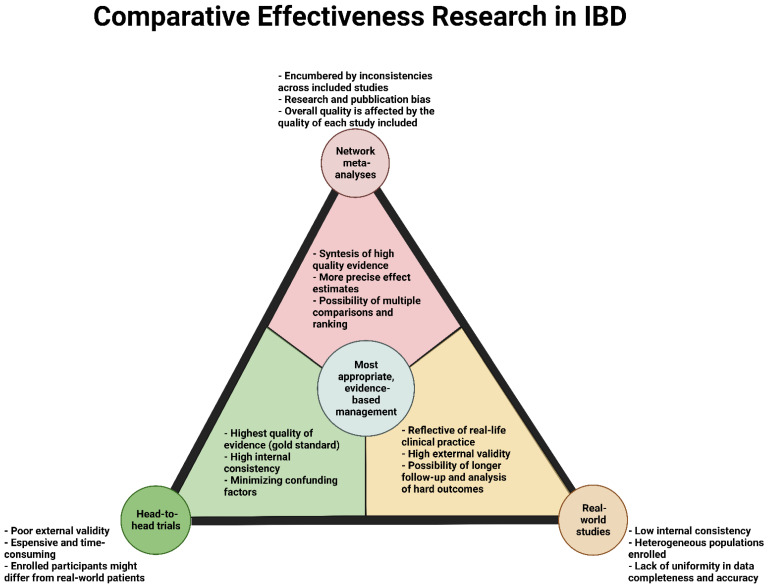
The 3 cornerstones of Comparative Effectiveness Research in IBD.

**Figure 4 jcm-11-06717-f004:**
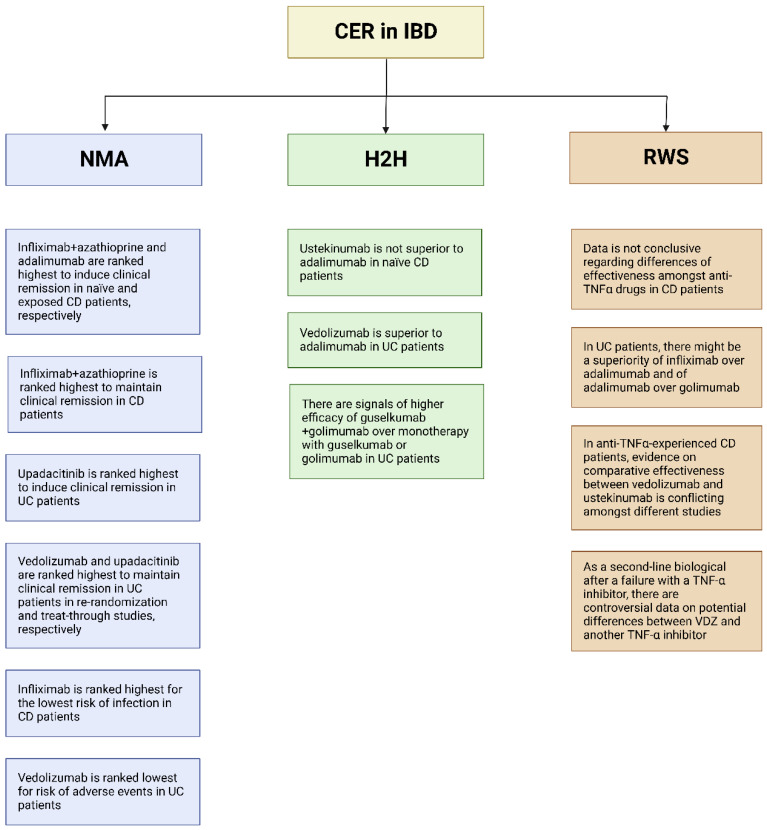
Main clinically relevant findings from CER in IBD.

**Table 1 jcm-11-06717-t001:** Summary of findings from the NMA by Singh et al. on the efficacy and safety of biologics in CD.

Induction of clinical remission—naïve patients
Pairwise meta-analysis: all treatments but certolizumab pegol are superior to placebo NMA: infliximab, infliximab + azathioprine, adalimumab and ustekinumab are superior to certolizumab pegol (moderate confidence in estimates) infliximab + azathioprine is superior to vedolizumab and certolizumab pegol (low confidence) infliximab + azathioprine and infliximab monotherapy are ranked first and second (SUCRA 0.96 and 0.81, respectively)
Induction of clinical remission—exposed patients
Pairwise meta-analysis: adalimumab, ustekinumab and risankizumab are superior to placebo vedolizumab is superior to placebo only to induce clinical response NMA: risankizumab and adalimumab are superior to vedolizumab (moderate and low confidence, respectively) adalimumab is ranked first (SUCRA 0.92), followed by risankizumab (SUCRA 0.77)
Maintenance of clinical remission—combined naïve and exposed patients
Pairwise: all agents are superior to placebo NMA: no significant differences among treatments infliximab + azathioprine and adalimumab are ranked first and second (SUCRA 0.87 and 0.78, respectively).
Safety—only from maintenance trials
adalimumab has a higher risk of infections compared to infliximab, infliximab+azathioprine, ustekinumab and placebo infliximab monotherapy has a higher risk of serious adverse event compared to infliximab+azathioprine infliximab + azathioprine, adalimumab and ustekinumab are ranked highest for the lowest risk of serious adverse events (SUCRA 0.86, 0.78 and 0.57, respectively) infliximab, infliximab + azathioprine and ustekinumab are ranked highest for the lowest risk of infections (SUCRA 0.90, 0.77 and 0.55, respectively).

**Table 2 jcm-11-06717-t002:** Summary of findings from the NMA by Lasa et al. on the efficacy and safety of biologics in UC.

UC
Induction of clinical remission
Pairwise meta-analysis: all interventions but filgotinib 100 mg are superior to placebo NMA: infliximab is superior to adalimumab, filgotinib 100 mg and 200 mg (moderate confidence in estimates), and to etrolizumab (high confidence) ozanimod, tofacitinib and ustekinumab are superior to filgotinib 100 mg (moderate confidence) upadacitinib is superior to all other agents (moderate-to-high confidence) upadacitinib and infliximab are ranked first and second (SUCRA 0.996 and 0.771, respectively).
Maintenance of clinical remission–treat-through studies
Pairwise meta-analysis: all agents are superior to placebo, vedolizumab is superior to adalimumab NMA: vedolizumab is superior to adalimumab (moderate confidence) vedolizumab and subcutaneous infliximab are ranked highest (SUCRA 0.906 and 0.716, respectively)
Maintenance of clinical remission–re-randomization studies
Pairwise meta-analysis: golimumab, etrolizumab and filgotinib 100 mg are not superior to placebo NMA: upadacitinib (both 15 mg and 30 mg) are superior to filgotinib 100 mg, ustekinumab, golimumab, and etrolizumab (moderate confidence) upadacitinib 30 mg is superior to ozanimod (moderate confidence) upadacitinib 30 mg is ranked highest (SUCRA 0.954), followed by upadacitinib 15 mg and filgotinib 200 mg (SUCRA 0.801 and 0.736, respectively).
Induction of endoscopic improvement
Pairwise meta-analysis: all agents but filgotinib 100 mg are superior to placebo. NMA: infliximab is superior to adalimumab, golimumab and vedolizumab (moderate confidence), and to etrolizumab (high confidence) ozanimod and tofacitinib are superior to adalimumab (moderate confidence) upadacitinib is superior to all other agents (moderate-to-high confidence) upadacitinib and infliximab are ranked first and second (SUCRA 0.999 and 0.783, respectively).
Maintenance of endoscopic improvement–treat-through studies
Pairwise meta-analysis: each treatment is superior to placebo NMA: vedolizumab is superior to adalimumab (moderate confidence) vedolizumab is ranked highest (SUCRA 0.967), followed by infliximab (SUCRA 0.676)
Maintenance of endoscopic improvement–re-randomization studies
Pairwise meta-analysis: filgotinib 100 mg is not superior to placebo NMA: upadacitinib 30 mg is superior to all other treatment, except for upadacitinib 15 mg (moderate confidence) upadacitinib 15 mg is significantly superior to ozanimod, filgotinib 100 mg, ustekinumab, golimumab and etrolizumab (moderate confidence) upadacitinib 30 mg, upadacitinib 15 mg and vedolizumab are ranked first to third (SUCRA 0.987, 0.861 and 0.718, respectively)
Induction of clinical response–naïve patients
Pairwise meta-analysis: all treatments but filgotinib 100 mg are superior to placebo NMA: infliximab is superior to adalimumab, etrolizumab, filgotinib 100 mg and filgotinib 200 mg (moderate confidence) ustekinumab, vedolizumab, golimumab, and ozanimod are superior to filgotinib 100 mg (moderate confidence) infliximab and ozanimod are ranked first and second (SUCRA 0.853 and 0.847, respectively)
Induction of clinical response–exposed patients
Pairwise meta-analysis: ustekinumab, tofacitinib, and filgotinib 200 mg are superior to placebo NMA: tofacitinib and ustekinumab are superior to adalimumab and vedolizumab (moderate confidence), and etrolizumab is superior only to adalimumab (moderate confidence) tofacitinib is ranked highest (SUCRA 0.927), followed by ustekinumab (0.887).
Induction of endoscopic improvement–naïve patients
Pairwise meta-analysis: all treatments but filgotinib 100 mg are superior to placebo NMA: infliximab is superior to adalimumab, golimumab and etrolizumab (moderate confidence) ustekinumab and ozanimod are superior to adalimumab (moderate confidence) ustekinumab is ranked highest in naïve patients (SUCRA 0.825), followed by ozanimod (SUCRA 0.798), filgotinib 200 mg (SUCRA 0.797) and then infliximab (SUCRA 0.753)
Induction of endoscopic improvement–exposed patients
Pairwise meta-analysis: only ustekinumab and tofacitinib are superior to placebo NMA: tofacitinib and ustekinumab are superior to both vedolizumab and adalimumab (moderate confidence) tofacitinib and ustekinumab are ranked first and second (SUCRA 0.936 and 0.851, respectively)
Safety
no differences in terms of risk of adverse events and serious adverse events are observed among treatments vedolizumab and golimumab have a lower risk of serious adverse events compared to placebo. upadacitinib ranked first for the highest risk of adverse events (SUCRA 0.843), followed by ustekinumab (SUCRA 0.697) ozanimod, placebo and etrolizumab ranked first to third for the highest risk of serious adverse events (SUCRA 0.831, 0.784 and 0.766, respectively) vedolizumab ranked lowest for the highest risk of adverse events and serious adverse events (SUCRA 0.184 and 0.139, respectively)

**Table 3 jcm-11-06717-t003:** Examples of different head-to-head (H2H) trial designs in IBD.

Author	Drugs	Disease/pts	Study Design	Primary End-Point	Main Finding
*Placebo-controlled trial with a non-powered reference arm*					
Kamm et al. [28]	MMX mesalazine vs. 5-ASA vs. placebo	343 UC pts	Phase 3 double-blind, double dummy parallel group, randomized, placebo-controlled trial	Remission rate (defined as a reduction of UC Activity index (DAI) <1 and at least 1 point of reduction in sigmoidoscopy score from the baseline	MMX mesalamine was efficacious and well-tolerated for the induction of clinical and endoscopic remission.
Sandborn et al. (CORE I) [29]	MMX budesonide 6 mg/die vs. 9 mg/die vs. 5-ASA vs. PL	509 UC pts	Phase 3, multicenter, randomized, double-blind, double-dummy, placebo-controlled trial	Remission rate (defined as a reduction of UC Activity index (DAI) <1 and at least 1 point of reduction in sigmoidoscopy score from the baseline	Budesonide MMX (9 mg) was safe and more effective than pl in inducing remission.
*Non-inferiority trials*					
Williams et al. (CONSTRUCT) [30]	Ciclosporin vs. infliximab	Acute severe ulcerative colitis refractory to IV CS	Single-blind	Quality- adjusted survival	No difference between ciclosporin and infliximab
Jørgensen et al. (Nor-switch) [31]	Infliximab originator vs. infliximab biosimilar (CT-P13)	482 pts including CD-UC or other IMID, at least 6 months stable infliximab treatment	Double-blind, randomized trials	Disease worsening	Switching from originator to biosimilar infliximab non- inferior to continuation of infliximab originator
*Superiority trials*					
Sands et al. (VARSITY) [23]	Vedolizumab IV vs. adalimumab SC	769 Moderate- to severe Active UC	Phase 3b, double-blind, double-dummy, randomized, active-controlled study	Achieving clinical remission	Vedolizumab superior to adalimumab
Sands et al. (SEAVUE) [16]	Adalimumab. vs. Ustekinumab	moderate-to-severely active CD	Phase 3b randomised, double-blind, parallel group, active-comparator	for clinical remission at week 5	Ustekinumab not superior to adalimumab at week 52
Danese S. (GARDENIA) [24]	Etrolizumab vs. infliximab	moderate-to-severely active UC patients naïve to TNF alpha inhibitor	Phase 3 randomised, double-blind, double-dummy, parallel-group,	Clinical response and Clinical remission	Primary end-point not reached
Sands et al. (VEGA) [32]	Golimumab + guselkumab vs. golimumab or guselkumab,	Moderately to Severely Active UC	Phase 2a Randomized, Double-Blind, Active-Controlled, Parallel- Group, Multicenter, Proof-of-Concept Study	Clinical responce	Combination significantly superior to golimumab monotherapy

Abbreviations: 5-ASA, 5-amminosalycilic acid; CD: Crohn’s disease; IV: intravenous; MMX: multimatrix; SC: subcutaneous; UC: ulcerative colitis.

**Table 4 jcm-11-06717-t004:** Comparative H2H ongoing trials in IBD according to Clinicaltrials.gov database.

Study Name or Number	Drugs	Disease	Phase	Design
INTREPID (NCT03759288)	Brazikumab vs. placebo and vs. adalimumab	Moderately to severely active CD	Phase 2 Phase 3	Multicenter, randomized, double-blind, placebo- and active-controlled
VIVID-1 (LY3074828)	Mirikizumab vs. ustekinumab	Moderately to severely active CD	Phase 3	Multicenter, randomized, double-blind, active-controlled
GALAXY (2–3)	Guselkumab vs. placebo and vs. ustekinumab	Moderately to severely active CD	Phase 2/3	Multicenter, randomized, double-blind, placebo- and active-controlled
TRIDENT (NCT04524611)	JnJ-64304500 vs. placebo and vs. ustekinumab	Moderately to severely active CD	Phase 2b	Multicenter, randomized, double-blind, placebo-controlled, parallel group
SEQUENCE (NCT04524611)	Risankizumab vs. ustekinumab	Moderately to severely active CD who have failed anti-TNFα therapy	Phase 3	Multicenter, randomized, efficacy assessor-blinded
NCT03558152	UTTR1147A vs. placebo and vs. vedolizumab	Moderately to severely active UC	Phase 2	Multicenter, randomized, parallel-group, double-blind, double-dummy, placebo-controlled

Abbreviations: CD, Crohn’s disease; UC, ulcerative colitis.

**Table 5 jcm-11-06717-t005:** Examples of different comparative real-world studies in IBD.

Author	Disease Drugs	Patients Followed	Main Outcomes	Results
Singh et al. [44]	CD ADA vs. IFX	2908 naïve pts	CD-related hospitalization Major abdominal surgery All-cause hospitalization	No significant differences (HR 0.81, 95% CI, 0.55–1.20) No significant differences (HR 1.24, 95% CI, 0.66–2.33) ADA was associated with reduced risk compared to IFX (HR 0.74, 95% CI, 0.56–0.97)
Macaluso et al. [45]	CD ADA vs. IFX	632 ptsNaïve ptsExperienced pts	Clinical benefit ^a^ at week 12 and 1 year	Week 12: No significant differences (OR 1.23, 95% CI, 0.63–2.44) 1 Year: No significant differences (OR 1.10, 95% CI, 0.61–1.96) Week 12: No significant differences (OR 0.72, 95% CI, 0.21–2.44) 1 Year: No significant differences (OR 1.23, 95%CI, 0.54–2.86)
Singh et al. [46]	UC ADA vs. IFX	1719 naïve pts	All-cause hospitalization Major abdominal surgery UC-related hospitalization Infections requiring hospitalization	ADA was associated with an increased risk compared to IFX (HR 1.84, 95% CI, 1.18–2.85) No significant differences (HR 1.35, 95% CI, 0.62–2.94) No significant differences (HR 1.71, 95% CI, 0.95–3.07) ADA was associated with an increased risk (HR 5.11, 95%CI, 1.20–21.8)
Renna et al. [47]	UC ADA vs. GOL	118 pts (both naïve and experienced)	Clinical benefit ^a^ at week 8 Discontinuation at end of FUP	ADA was superior to GOLI (HR 2.88, 95%CI 1.31–633) ADA was associated with a lower risk compared to GOLI (HR 0.49, 95% CI 0.30–0.81)
Bohm et al. [48]	CD VDZ vs. Anti-TNF-α	1266 (both naïve and experienced)	Incidence of serious infections Incidence of SAE Incidence of non-infectious SAE	No significant differences (OR 1.18, 95% CI 0.78–1.79) No significant differences (OR, 0.751, 95% CI 0.51–1.08) VDZ was superior to anti-TNF-α drugs (OR, 0.07, 95% CI 0.01–0.24)
Lukin et al. [49]	UC VDZ vs. Anti-TNF-α	722 pts (both naïve and experienced)	Incidence of serious infections Incidence of SAE	No significant differences (HR 1.23, 95% CI, 0.60–2.51) No significant differences (HR 0.89; 95% CI 0.50–1.61)
Alric et al. [50]	CD UST vs. VDZ	239 pts (107 vs. 132)	Clinical remission at week 48 SFCR at week 48 Persistence on therapy at week 48	UST was superior to VDZ (OR 1.92, 95% CI, 1.09–3.39) No significant differences (OR 1.57, 95%CI, 0.88–2.79) UST was superior to VDZ (OR 2.54, 95%CI, 1.40–4.62)
Townsend et al. [51]	CD UST vs. VDZ	130 pts (45 vs. 85)	SFCR at 2 months SFCR at 12 months	UST was superior to VDZ (OR 2.79, 95%CI, 1.06–7.39) Uste was superior to Vedo (OR 2.01, 95%CI 0.89–4.56)
Biemans et al. [52]	CD UST vs. VDZ	213 pts (85 vs. 128)	SFCR at week 52 Biochemical remission at week-52 Combined SFCR + biochemical remission at week 52 Safety outcomes ^b^	UST was superior to VDZ (OR 2.58, 95% CI, 1.36–4.90) UST was superior to VDZ (OR 2.34, 95% CI, 1.10–4.96) UST was superior to VDZ (OR 2.74, 95% CI, 1.23–6.09) No significant differences for each outcome
Onali et al. [53]	CD UST vs. VDZ	470 pts (239 vs. 231)	Clinical response at week 26 Clinical remission at week 52 SFCR at week 52 Objective clinical response/remission at week 52	No significant differences (OR 1.26, 95% CI, 0.83–1.90) VDZ was superior to UST (OR 1.69, 95% CI, 1.13–2.52) VDZ was superior to UST (OR 1.53, 95%, CI 1–02–2.28) No significant differences for both comparisons
Lenti et al. [54]	CD UST vs. VDZ	393 pts (275 vs. 118)	Clinical remission at week 14 Clinical remission at week 52	UST was superior to VDZ (38% of increased likelihood, 95% CI, 25–50, *p* < 0.001) No significant differences (9% of increased likelihood, 95% CI, 15–33, *p* = 0.462)
Manlay et al. [55]	CD UST vs. VDZ	312 pts (224 vs. 88)	SFCR at week 54 Deep remission at week 14	UST was superior to VDZ (50.6% vs. 40.6%, *p* = 0.047) UST was superior to VDZ (17.9% vs. 5.7%, *p* = 0.047)

Abbreviations: ADA: adalimumab; CD: Crohn’s disease; CI: Confidence Interval; FUP: Follow-up; GOL: golimumab; HR: Hazard Ratio; IFX: infliximab; OR: Odds Ratio; pts: patients; SAE: serious adverse events; SFCR: Steroid-free clinical remission; UST: ustekinumab; VDZ: vedolizumab. ^a^ Clinical benefit: sum of steroid-free clinical remission plus clinical response. ^b^ Rate of infections, adverse events, and hospitalization.

## Data Availability

Not applicable.

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
