# Peer review of "Comparative Effectiveness Research: A Roadmap to Sail the Seas of IBD Therapies"

_jcm, 2022, doi:10.3390/jcm11226717_

Round 1

Reviewer 1 Report

The authors in their review Comparative Effectiveness Research: a roadmap to sail the seas of IBD therapies have summarised the subject nicely. It is a timely review. I have a few suggestions.

Major points

1.      The authors should add key messages from each methodology (NMA, H2H, RWS) regarding the current positioning of the therapies. As is done for the two NMAs in Tables 1 and 2; a summary tables of H2H and RWS would add to the readability.

2.      Data on microbiome targeted therapies such as FMT and dietary therapies has not been discussed. Please refer to these as well. The following references can be of help

a.      Zhou HY, Guo B, Lufumpa E, Li XM, Chen LH, Meng X, Li BZ. Comparative of the Effectiveness and Safety of Biological Agents, Tofacitinib, and Fecal Microbiota Transplantation in Ulcerative Colitis: Systematic Review and Network Meta-Analysis. Immunol Invest. 2021 May;50(4):323-337. doi: 10.1080/08820139.2020.1714650. Epub 2020 Feb 2. PMID: 32009472.

b.      Vuyyuru SK, Kedia S, Kalaivani M, Sahu P, Kante B, Kumar P, Ranjan MK, Makharia G, Ananthakrishnan A, Ahuja V. Efficacy and safety of fecal transplantation versus targeted therapies in ulcerative colitis: network meta-analysis. Future Microbiol. 2021 Oct;16:1215-1227. doi: 10.2217/fmb-2020-0242. Epub 2021 Sep 30. PMID: 34590904.

3.      It would also be prudent to discuss corticosteroids and immunesuppressants and not just restrict the discussion to biologics.

4.      It is important to discuss that the literature referenced to in the NMAs and H2H trials exclude patients with mild UC and fistulising/structuring CD. This is a limitation of the available data.

5.      A conclusion image/table at the end depicting the current positioning of the therapies based on the current results of the CER is suggested.

Minor points

1.      Table 1. Summary of NMA by Singh S et al.

a.      It is mentioned that adalimumab has a higher risk of infections compared to infliximab, infliximab + azathioprine, ustekinumab and placebo. Please specify that this is in trials on maintenance of remission.

b.      Infliximab, infliximab + azathioprine and ustekinumab are ranked highest for the lowest risk of infections (SUCRA 0.90, 0.77 and 0.55,respectively).The data presented is not in sync with the NMA.

2.      English language and grammar needs editing.

a.      Certain spellings need to be corrected. For example in Table 2, is is written as “si”. Line 228 replace “e” with “and”.

b.      Lines 226, 234, 235 : replace “die” with “day”

c.      The grammar and punctuation marks need to be corrected. For example lines 36-40, 49-53, etc.

d.      Line 66. Correct spelling of “with”

e.      Line 68 : effectiveness “is used” to describe…

3.      Add reference to the definition of CER.

4.      Headings 3.1 and 3.2 are same. Please see.

Author Response

REVIEWER 1

Major points

  1. The authors should add key messages from each methodology (NMA, H2H, RWS) regarding the current positioning of the therapies. As is done for the two NMAs in Tables 1 and 2; a summary tables of H2H and RWS would add to the readability.

Author’s reply: We appreciate the suggestion. We have arranged two tables that summarize the main findings from H2H and RW studies (tables 3 and 5, respectively). Furthermore, there is now a new figure (Figure 4) that presents the most clinically relevant messages, in the IBD field, from each methodology.

  1. Data on microbiome targeted therapies such as FMT and dietary therapies has not been discussed. Please refer to these as well. The following references can be of help
  2. Zhou HY, Guo B, Lufumpa E, Li XM, Chen LH, Meng X, Li BZ. Comparative of the Effectiveness and Safety of Biological Agents, Tofacitinib, and Fecal Microbiota Transplantation in Ulcerative Colitis: Systematic Review and Network Meta-Analysis. Immunol Invest. 2021 May;50(4):323-337. doi: 10.1080/08820139.2020.1714650. Epub 2020 Feb 2. PMID: 32009472.
  3. Vuyyuru SK, Kedia S, Kalaivani M, Sahu P, Kante B, Kumar P, Ranjan MK, Makharia G, Ananthakrishnan A, Ahuja V. Efficacy and safety of fecal transplantation versus targeted therapies in ulcerative colitis: network meta-analysis. Future Microbiol. 2021 Oct;16:1215-1227. doi: 10.2217/fmb-2020-0242. Epub 2021 Sep 30. PMID: 34590904.
  4. It would also be prudent to discuss corticosteroids and immunesuppressants and not just restrict the discussion to biologics.

Author’s reply for comments 2 and 3: We thank the Reviewer for their suggestion. However, we believe that adding more data on FMT, corticosteroids and immunosuppressors would not be consistent with the purpose of this present article. Indeed, our primary goal, with this paper, was to discuss the main characteristics and limitations of each different methodology that is used in comparative effectiveness research (i.e., network meta-analyses, head to head trials and real-life comparisons) in IBD; we chose to focus exclusively on biologics and small molecules, as their positioning is one of the main issues in the IBD field at the present moment; hence, we attempt to provide some useful examples on how to interpret data coming from studies on this topic. By including data also on FMT, corticosteroids and immunosuppressors, we would probably end up increasing the length of the manuscript to a point where readability would be compromised. Furthermore, it would go beyond our scope, as we do not aim, in this article, to provide a comprehensive overview nor a decalogue on IBD therapies, but just to set a framework that could help physicians in interpreting data coming from different sources of evidence-based medicine. However, we agree with the Reviewer that mentioning the possibility of comparing biologics/small molecules with other treatment strategies is important: thus, we included a mention of that in the ‘Conclusion’ paragraph (cf. lines 583-587).

  1. It is important to discuss that the literature referenced to in the NMAs and H2H trials exclude patients with mild UC and fistulising/structuring CD. This is a limitation of the available data.

Author’s reply: We appreciate the input. We have included such considerations in the appropriate paragraphs (cf. lines 211-216, 231-234).

  1. A conclusion image/table at the end depicting the current positioning of the therapies based on the current results of the CER is suggested.

Author’s reply: As suggested, we have prepared such a figure (Figure 4) that is now presented and discussed in the ‘Conclusion’ paragraph.

Minor points

  1. Table 1. Summary of NMA by Singh S et al.
  2. It is mentioned that adalimumab has a higher risk of infections compared to infliximab, infliximab + azathioprine, ustekinumab and placebo. Please specify that this is in trials on maintenance of remission.
  3. Infliximab, infliximab + azathioprine and ustekinumab are ranked highest for the lowest risk of infections (SUCRA 0.90, 0.77 and 0.55,respectively).The data presented is not in sync with the NMA.

Author’s reply: In regard to ‘point a’, all safety data are referred to maintenance trials (as it is now specified in the table), as the authors of the NMA did not perform analysis on safety outcomes from induction studies. In regard to ‘point b’, we believe that our statement reflects the data presented in the NMA (the exact quote is: “for the risk of infection, infliximab combination therapy with azathioprine (SUCRA 0·90), infliximab alone (SUCRA 0·77), and ustekinumab (SUCRA 0·55) were highest rated (lowest risk of infection), followed by vedolizumab (SUCRA 0·41), adalimumab (SUCRA 0·16), and certolizumab pegol (SUCRA 0·10)”).

  1. English language and grammar needs editing.
  2. Certain spellings need to be corrected. For example in Table 2, is is written as “si”. Line 228 replace “e” with “and”.
  3. Lines 226, 234, 235 : replace “die” with “day”
  4. The grammar and punctuation marks need to be corrected. For example lines 36-40, 49-53, etc.
  5. Line 66. Correct spelling of “with”
  6. Line 68 : effectiveness “is used” to describe…

Author’s reply: We went through the manuscript and corrected the typos we identified.

  1. Add reference to the definition of CER.

Author’s reply: The definition is already referenced: reference 7 belongs to the whole sentence included within the quotation marks (cf. lines 56-61).

  1. Headings 3.1 and 3.2 are same. Please see.

Author’s reply: We corrected it.

Reviewer 2 Report

The authors are to be commended on addressing this important and relevant topic. There are however a few issues with the article as it stands

1. The article is too long and lacks focus. The authors have tried to cover too much ground which makes it challenging to read. The article needs to be shortened and more concise

2. While the article centers around NMA, HTH trials, and RWE, these sections can be condensed

3. For the reader unfamiliar with SUCRA the authors should clarify what these values mean

4. The article needs to be more clinically relevant in order to assist the practicing Gastroenterologist in choosing advanced therapies. The authors should attempt to summarize all the evidence and try and make some recommendations at the end of the article on how to apply this data in clinical practice, appreciating the challenges therein

5. I am not sure if discussing etrolizumab is of any value given that it will likely never enter clinical practice

6. The authors discuss trials such as the MMX mesalamine and budesonide trials, CONSTRUCT, Nor-switch, and VEGA, to illustrate the different types of HTH trials. This muddies the water with too much information and takes focus away from the real issue at hand - what 1st and 2nd line advanced therapies should we be using in 2022? It would suffice to provide a brief overview of these study designs without giving detailed examples

7. The conclusion does not sum up the article content as there is no mention of the IBD component

8. Figure 3 and reference 55 do not belong in the conclusion but should rather be in the text

Author Response

REVIEWER 2

  1. The article is too long and lacks focus. The authors have tried to cover too much ground which makes it challenging to read. The article needs to be shortened and more concise
  2. While the article centers around NMA, HTH trials, and RWE, these sections can be condensed

Author’s reply to Comments 1 and 2: We see the Reviewer’s point. Our main goal, in this paper, is to delineate the main characteristics and limitations of the different methodologies that are used in comparative research in the IBD field, rather that providing an extensive and comprehensive overview on the available literature. Hence, we shortened the text by preparing summarizing tables for most of the findings from different studies (Tables 3 and 5), thereby leaving in the main text specific comments on such findings and their interpretation. 

  1. For the reader unfamiliar with SUCRA the authors should clarify what these values mean

Author’s reply: An additional specification on the interpretation of SUCRA values has been included in the text (cf. lines 134-137).

  1. The article needs to be more clinically relevant in order to assist the practicing Gastroenterologist in choosing advanced therapies. The authors should attempt to summarize all the evidence and try and make some recommendations at the end of the article on how to apply this data in clinical practice, appreciating the challenges therein

Author’s reply: We understand the point raised by the Reviewer, but we kindly disagree with them. Recommendations are already available in the guidelines published by scientific societies, and they are based on the results of comparative effectiveness research. Our recommendations would risk to just reiterate those statements – with the additional flaw that they would lack a likewise sound methodology, as we did not perform a systematic review of all available evidence (as this would go beyond our scope) nor we could apply the GRADE method to our recommendations. On the contrary, we believe that the primary value of our work for gastroenterologists is represented by the fact that it is explicative in regard to the different methodologies that inform comparative effectiveness research, hence it helps to critically interpret CER studies and society guidelines and how this evidence can be transported into everyday clinical practice. However, to increase the practical utility of this present paper, we included ad additional figure (Figure 5) that summarizes the most important findings derived from each CER methodology in IBD. Of note, as opposed to guidelines, we also presented data on real-world studies, which are undoubtedly useful to provide information on the effectiveness of different therapies as well as on patients’ populations that are most likely to benefit from them.

  1. I am not sure if discussing etrolizumab is of any value given that it will likely never enter clinical practice

Author’s reply: Thanks for the suggestion. We decided to discuss data on etrolizumab because, although the drug will probably not be licensed as a therapy for IBD, the related trial served as an example of a methodology in which an incorrect design leads to the difficult interpretation of the data.

  1. The authors discuss trials such as the MMX mesalamine and budesonide trials, CONSTRUCT, Nor-switch, and VEGA, to illustrate the different types of HTH trials. This muddies the water with too much information and takes focus away from the real issue at hand - what 1st and 2nd line advanced therapies should we be using in 2022? It would suffice to provide a brief overview of these study designs without giving detailed examples

Author’s reply: Thanks for the suggestion. The reason why we included these studies was to provide examples of different designs of H2H trials in IBD and comment on potential strengths and limitations. To increase the readability of the paper, we summarized the findings coming from these studies in Table 3, leaving in the text just the comments on the trials designs and the conclusions that can be drawn from those data.

  1. The conclusion does not sum up the article content as there is no mention of the IBD component

Author’s reply: Following the Reviewer’s input, we re-arranged the ‘Conclusion’ paragraph. It now includes Figure 4, which summarizes the most clinically relevant findings originating from different CER methodologies; furthermore, we comment on the limited utility of currently available evidence, and on the subsequent need for further, methodologically solid research.

  1. Figure 3 and reference 55 do not belong in the conclusion but should rather be in the text

Author’s reply: As suggested, we moved reference 55 in the NMA paragraph. However, we believe that figure 3 does belong to the ‘Conclusion’ paragraph, as it serves the purpose of summarizing all the methodological features associated with each source of evidence in CER, as they are discussed throughout the paper.

Round 2

Reviewer 1 Report

The authors have addressed the concerns appropriately.

I have a few minor suggestions

 1.      The SEAVUE, VARSITY, GARDENIA and VEGA are head to head comparative trials but not superiority trials. “Superiority trials” is a technical term with well-defined superiority margins which was not the design of these studies. Would recommend removing this term from the table as well as the text and summarise the studies under a common head of head to head trials.

2.      Similarly CONSTRUCT is not a non-inferiority study. It is a mixed methods pragmatic randomised trial. Please correct this.

3.      Please mention that the review focuses only on biologics and small molecules and not on other available therapies in the abstract as well.

Author Response

We thank the Reviewer for their time and effort. In regard to their latest suggestions, we see their point on the study designs of head-to-head trials, but we kindly disagree with them. Specifically:

1- SEAVUE, VARSITY, GARDENIA and VEGA are defined by the authors themselves as superiority trials, and they do include the necessary “delta” in the statistical calculation. Please, see in material and methods, statistical analysis section (e.g., SEAVUE: “ Statistical analysis: The study was powered to evaluate ustekinumab superiority over adalimumab for clinical remission ….to detect a 15% difference between the ustekinumab and adalimumab groups….”).

2- The CONSTRUCT trial is mixed methods, open-label, parallel-group, pragmatic randomised trial, designed by the authors as a non-inferiority trial. In fact, they do not include a significant delta between the two arms in the sample size calculation. Please, see the statistical analysis section: “Our hypothesis was that there is no difference in the clinical effectiveness of these two treatments…….Our original target sample size was 360 participants with analysable data, based on an equivalence design, an effect size of 0·30…).

As suggested, we have modified the abstract, where we specify that the review primarily focuses on the findings on biologics and small molecules.